# Navigating the Grey Area: How Expressions of Uncertainty and Overconfidence Affect Language Models

**Kaitlyn Zhou**
Stanford University
katezhou@stanford.edu

**Dan Jurafsky**
Stanford University
jurafsky@stanford.edu

**Tatsunori Hashimoto**
Stanford University
thashim@stanford.edu

## Abstract

The increased deployment of LMs for real-world tasks involving knowledge and facts makes it important to understand model epistemology: what LMs think they know, and how their attitudes toward that knowledge are affected by language use in their inputs. Here, we study an aspect of model epistemology: how *epistemic markers* of certainty, uncertainty, or evidentiality like *"I'm sure it's"*, *"I think it's"*, or *"Wikipedia says it's"* affect models, and whether they contribute to model failures. We develop a typology of epistemic markers and inject 50 markers into prompts for question answering. We find that LMs are highly sensitive to epistemic markers in prompts, with accuracies varying more than 80%. Surprisingly, we find that expressions of high certainty result in a 7% decrease in accuracy as compared to low certainty expressions; similarly, factive verbs hurt performance, while evidentials benefit performance. Our analysis of a popular pretraining dataset shows that these markers of uncertainty are associated with answers on question-answering websites, while markers of certainty are associated with questions. These associations may suggest that the behavior of LMs is based on mimicking observed language use, rather than truly reflecting epistemic uncertainty.

## 1 Introduction

As natural language systems are increasingly used in situations involving factuality and knowledge, it becomes important for LMs to be able to interpret how humans talk about knowledge. LMs must learn to accurately interpret linguistic cues like *expressions of uncertainty and certainty* that are used to talk about confidence, source, and limitations of information. In this work, we seek to understand how models interpret this linguistic phenomenon by measuring how language generation varies when prompted with expressions of uncertainty.

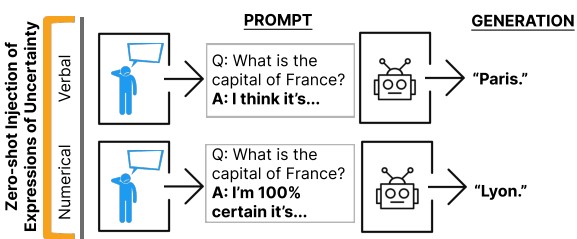

Figure 1: Using zero-shot promoting to inject verbal and numerical uncertainties into trivia questions. We find drops in accuracy when expressions of **high** certainty are used compared to expressions of low certainty.

Naturalistic expressions of uncertainty/certainty cover a broad range of discourse acts such as signaling hesitancy, attributing information, or acknowledging limitations. Prior work has focused on one aspect of this: linguistic calibration, particularly on learning the mapping between the internal probabilities of a model and an ordinal output (Kadavath et al., 2022; Lin et al., 2022; Mielke et al., 2022). Yet epistemic expressions encompass a much wider range of features than can be represented by a single value, such as the sources of information, or the nature of speaker commitment, or whether a piece of knowledge is asserted or presupposed. Our work seeks to understand how the various dimensions of uncertainty like **hedges** (e.g., *"It could be..."*), **factive verbs** (e.g., *"We realize it's..."*), and **evidential markers** (e.g., *"Wikipedia says it's..."*) impact language generations. By shifting our focus to naturalistic expressions of uncertainty and certainty, we would enable models to flexibly interpret a wider range of uncertainties otherwise not possible under the current linguistic calibration setup.

To understand how epistemic expressions affect language models, we use zero-shot prompting and inject verbal and numerical markers into trivia question prompts. This process converts a prompt like "What is the capital of France?" to, "What is the capital of France, *I think it's...*". (Figure 1). Aided by our linguistic typology, we then measure how different epistemic marker types impact model ac-

curacy. By doing so, we complement current linguistic calibration work and paint a more complete picture of the role of these epistemic markers.

We begin with two broad hypotheses on how LMs might respond to expressions of uncertainty. First, we might suppose that models are robust to any added expressions of uncertainty in the prompt. An alternative hypothesis is that, models might respond differently based on the uncertainty cues, and using a marker suggesting certainty or confidence might be more likely to produce the correct response than a prompt with low certainty or confidence. Under the latter hypothesis, we might expect performance for a prompt with no epistemic markers (which we call the *standard method*) to lie in between these two. This second hypothesis would also be consistent with prior work showing that LMs can generate language in the style of diverse personas (Lee et al., 2022; Park et al., 2022).

Surprisingly, we find that injecting expressions of high certainty like *"I'm certain it's"* or *"I'm 100% sure"* actually leads to lower accuracy. We follow up with qualitative and quantitative analysis of popular training data, suggesting some potential sources of these unexpected behaviors. Our work thus offers three key contributions:

- We introduce a typology of expressions of uncertainty to evaluate how linguistic features impact LM generation.

- We demonstrate how model accuracy suffers when expressions of certainty (e.g., *"I'm certain"*) are used.

- We perform qualitative and quantitative analysis to reveal the potential origins of these unexpected behaviors.

Together, our findings illustrate the shortcomings of models' abilities to interpret expressions of uncertainty and highlight the gaps that still exist between natural and generated language.[1]

## 2  Expressions of Certainty and Uncertainty: Linguistic Background

A broad linguistic literature exists in epistemological markers related to certainty and uncertainty, both for strengthening/weakening category membership and strengthening/weakening speaker commitment to the truth value of a proposition (see Related Work). For convenience, we broadly

---

[1]Details: https://github.com/katezhou/navigating_the_grey

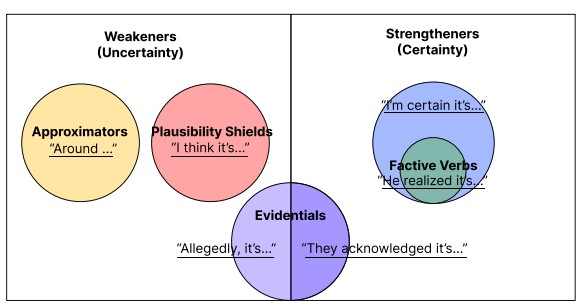

Figure 2: Lexical Features of Expressions of Uncertainty. Uncertainty classification partly adapted from (Prince et al., 1982). Certainty markers are strengtheners which contain factive verbs. Evidential markers can be both expressions of certainty and uncertainty (strengtheners or weakeners). Personal pronouns and reference to sources are additional dimensions of expressions of (un)certainty not shown in this diagram.

group these linguistic devices into **weakeners** and **strengtheners**.

The most widely studied weakeners are hedges, first defined by Lakoff (1975) as related to modifying or weakening the category membership of a predicate or nominal. The central kind of hedges are **approximators** (e.g., *somewhat*, *kind of*, *about*, *approximately*), which hedge propositional content. Another class of weakeners that some (like Prince et al. (1982)) but not others classify under hedges are **plausibility shields** which express a speaker's lower level of commitment (e.g., *I think*, *I believe*).

**Strengtheners** are constructions that mark *certainty*. We use this term to refer to strengthening speaker commitment to truth value, and also to strengthening category membership. Strengtheners include **boosters** or **intensifiers** (e.g., *"I am certain"*, *"Undoubtedly"*) (Hyland, 2005, 2014).

While boosters can *assert* certainty or truth, a second kind of strengthening construction, the **factive verb**, is used to *presuppose* certainty or truth. Factive verbs like *"know"*, *"realize"*, or *"understand"* (Kiparsky and Kiparsky, 1970) presuppose the truth of the complement sentence. The statement *"X realizes Y"* presupposes that Y is true, (and also asserts that X is aware of it). By contrast, *"X believes Y"* makes no presupposition about the truth of Y. Thus the sentence *"He realizes [Madrid is the capital of France]"* is infelicitous, because *"realize"* as a factive verb presupposes the truth of its complement clause, which in this case is false (Madrid is not the capital of France).

A third class of markers can mark both certainty and uncertainty by directly indicating the source of the information. Such expressions (e.g., *"Accord-*

ing to Wikipedia", "I heard", "I saw") are called **evidential markers**, linguistic signals that tells the hearer where the information came from (Aikhenvald, 2004). One subtype of evidential markers, quotative markers, are used when reported information overtly references a source (e.g., *"According to research in the latest issue of Nature"*, *"Two recent studies demonstrate that…"*). We examine when references are citing a source (e.g., *"Wikipedia says"*) versus when the source is unspecified or very indirect (e.g., *"They said"*). We'll refer to this former case as **sourced** as a shorthand for indicating that a source is mentioned.

Finally, first-person **personal pronouns** (e.g., *"I"*, *"we"*, *"our"*) can be used to mark subjectivity and uncertainty in expressions like "*I think*".

## 2.1 Expressions of Uncertainty Typology

Using a bottom-up process, we gathered a diverse set of templates guided by the linguistic literature to develop a single, coherent taxonomy and classify templates. To gather the templates, the authors relied on the literature from Prince et al. (1982), brainstormed additional templates, and used crowd sourcing via Amazon Mechanical Turk. The authors then drew on linguistic literature to augment this list, adding dimensions such as factive verbs (Kiparsky and Kiparsky, 1970), evidentials (a broader category of attributions) (Aikhenvald, 2004), and authoritative sources. Finally, they integrated this literature, developed a coherent taxonomy, and classified each of our templates. Figure 2 illustrates how these markers relate to certainty and uncertainty in our coding scheme.

The final list of templates includes: weakeners, strengtheners, plausibility shields, factives, evidential markers, mentions of sources, and personal pronouns (Appendix Table 6). Each expression is then coded for the linguistic features above, allowing us to analyze how epistemological modifiers impact language modeling in the QA setting.[2]

## 2.2 Amazon Mechanical Turk Details

We use Amazon Mechanical Turk to crowd-source additional expressions of uncertainty (Figure 7 in Appendix). Workers were filtered to be have HITs greater than 99 and to have at least 500 approved HITs. Given the simplicity of the task, we esti-

mated it would take users a minute or two to complete the task, a paid users $0.35 USD for the task which results in roughly $10.50 USD to $21.00 USD an hour. We collected a total of 9 samples of 5 examples each. The authors then read, filtered, and modified the examples to follow the overall linguistic structure of the other templates. This study did not require IRB approval as it is not a study of humans and we do not collect any information about the annotators themselves.

## 3 Methods

To study how models interpret uncertainty, we inject markers into trivia questions in open-ended question-answering. Using our typology, we create fifty sentences (minimal pairs) for every question.

Our datasets include TriviaQA, a standard QA dataset (Joshi et al., 2017); Natural Questions (closed-book), an aggregated set of Google queries (Kwiatkowski et al., 2019a); CountryQA, which we constructed using names of countries and capitals in the form of "What is the capital of Afghanistan?"; and Jeopardy questions crawled from a fan-created database, J! Archive.[3] We use a random subset of 200 questions for three of our datasets. For CountryQA, we used all 53 questions whose answers were in vocabulary (details in 3.2).

Although the total number of questions per template is small, we present our key findings in aggregate across all templates with shared characteristics, increasing our statistical power. We test each of our fifty templates across this subset of questions and calculate 95% confidence intervals using bootstrap resampling or $t$-tests.

We primarily study OpenAI's GPT models as they are commonly used large language models with reasonable performance on QA tasks (Liang et al., 2023). For all models, except GPT4, we set temperature to 1. For the generated tokens, we take the sum of the probability assigned to the gold answer(s) to be the *probability-on-gold*. When calculating accuracy, we generate 10 tokens and if any of the tokens match the answer (or aliases of the answer), we'll count that as a correct generation. This is done to not unfairly disadvantage templates that are prone to generating words prior to emitting the answer (e.g., "Allegedly, it's said to be…").[4] For GPT4, where the log probabilities were not

---

[2]Approximators, which are primarily used for expressing uncertainty of continuous items are excluded from our templates as we are primarily working with trivia questions with discrete answers.

[3]https://j-archive.com/
[4]Authors also manually inspected 1,000 answers to ensure this approach doesn't lead to false positives e.g., *"The answer is not Paris."*

available, we set the temperature to 0, also ensuring a (mostly) deterministic output, and used the generated text.

## 3.1 Additional Prompting Details

For Section 4, we use OpenAI's researcher API and retrieve the top 50 most probable predictions per token. For Section A.1 we use the standard API and retrieve the top 5 most probable predictions per token.

We are careful that our prompts do not end with trailing white space (" ") as recommended by OpenAI in order to prompt the best generations. We also use the delimiters "Q:" and "A:" to signal the start and end of questions and answers.[5]

Lastly, for additional transparency and given the frequent changes in models, we also provide timestamps of when the models were prompted. We used the Davinci model from Feb 2023 on all four datasets. Experiments on Ada, Babbage, Curie, text-davinci–03, and GPT4 models on the TriviaQA dataset were all conducted in June of 2023. Experiments on Ada, Babbage, Curie, text-davinci, and GPT4 models on the CountryQA, NaturalQA, and Jeopardy datasets were conducted the week of August 28th, 2023.

## 3.2 Challenges in Open-Ended Generation

A key challenge of open-ended generation is measuring log probabilities of generated tokens, needed for our analysis. Answers can be paraphrased in a myriad of ways, making the scoring of tokens difficult. Furthermore, some answers are OOV which necessitates multi-subtoken generation. To avoid these potential confounders, we filter out questions with multi-word or OOV answers, allowing us to more accurately measure the probability placed on the gold answers.

The filtering of questions has the added advantage of prioritizing questions with high-frequency answers. By focusing on questions with high-frequency answers, the resulting dataset contains questions for which LMs are more likely to know the right answer. This helps mitigate the effects of performance variation that may result from a wide range of question difficulty as well as demonstrate that our observed effects are not isolated to unusual or rare questions.

[5]In CountryQA and TriviaQA, an additional new line character was added before "A:". To reduce accruing additional costs, NaturalQA and Jeopardy results were not rerun to match this exact template.

## 4 The Impact of Uncertainty on Language Generation

Information in real-life text is rarely in black-and-white, and expressions of uncertainty are necessary in supporting decision-making processes. In this section, we investigate how GPT3's generation changes based on the verbal expressions of uncertainty in its prompt (Figure 1, top) and whether some expressions of uncertainty can have systematic effects on model behavior.

## 4.1 Variation in GPT3 Responses Across Verbal Uncertainties

We evaluate GPT3 (davinci) using zero-shot prompting on our four datasets. We find that the results do not support our first hypothesis: GPT3 is *highly* sensitive to uncertainty cues, and accuracy changes significantly depending on the certainty/uncertainty expression used. Across our datasets, we find that accuracies can change by up to 80% on the exact same set of questions. This is especially pronounced in CountryQA where a template like, *"We realize it's..."* achieves 14% accuracy while many other templates result in perfect accuracy. In TriviaQA, when prompted with a template such as *"I'm certain it's..."* the model's accuracy is 42% but when prompted with "*I would need to double check but maybe it's..."*, accuracy increases to 56%. Our findings illustrate that expressions of uncertainty affect language generation and that the changes resulting from these expressions have substantive impact on overall accuracy.

Turning to our second hypothesis, we seek to understand how GPT3's responses to QA answers change based on the expression of uncertainty used in the prompt. Surprisingly, we find that weakeners perform significantly better than strengtheners across all four of our datasets. The average accuracy among weakeners across all four datasets is 47% compared to 40% among strengtheners. This effect is especially large in CountryQA where the accuracy gap is 17%. This effect is driven by the use of factive verbs in strengtheners (as nearly all uses of factive verbs in our templates are strengtheners)[6], and the use of factive verbs consistently results in significant losses in accuracy (Figure 3). In other words, when the template presupposes the truth, accuracy drops.

This finding contradicts the second hypothesis, as we might have expected expressions of certainty

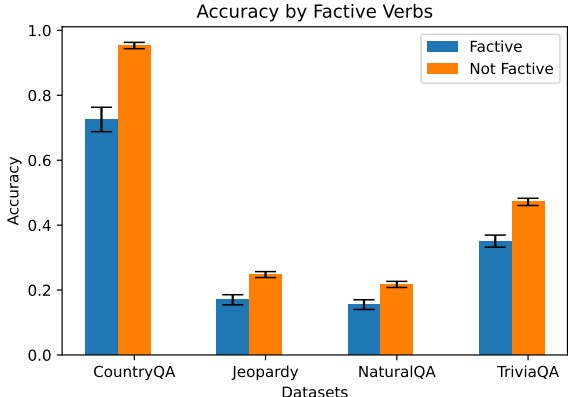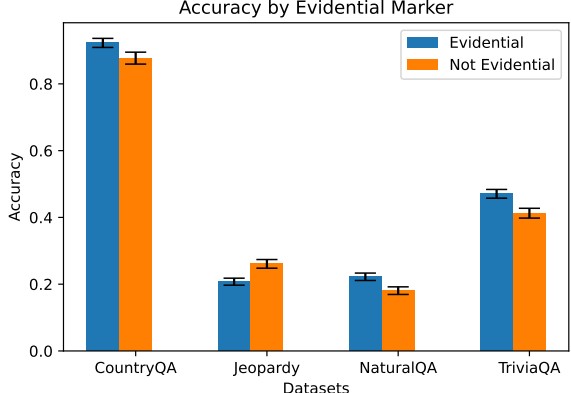

Figure 3: Significant and consistent accuracy losses for templates with factive verbs (left). Evidential markers significantly improves accuracy in three out of four datasets (right). 95% CI calculated using bootstrap resampling. Visualizing results for GPT-3 (davinci).

|  | ada | babbage | curie | davinci | instruct | gpt-4 |
|---|---|---|---|---|---|---|
| Boosters | 0.091 | 0.257 | 0.313 | 0.392 | 0.589 | 0.793 |
| Hedges | 0.079 | 0.272 | **0.333***** | **0.468***** | **0.642***** | **0.822***** |
| Factive Verbs | 0.078 | 0.237 | 0.293 | 0.347 | 0.555 | 0.771 |
| Non-Factives Verbs | **0.085*** | **0.276***** | **0.336***** | **0.468***** | **0.641***** | **0.821***** |
| Evidentials | **0.087**** | **0.281***** | **0.347***** | **0.449*** | **0.640***** | **0.820***** |
| Non-evidentials | 0.080 | 0.250 | 0.301 | 0.433 | 0.601 | 0.799 |

Table 1: Across all six models tested, hedges outperform boosters, non-factive verbs outperform factives and evidentials out-perform non-evidentials. (Instruct = `text-davinci-003`, GPT4 uses context window 32K.) $t$-test $p$-values, * < 0.05, ** < 0.01, *** < 0.001**.

to improve performance, not hurt it. This is particularly concerning as confident prompts —which LM users might naturally expect to result in better generations— actually lead to worse generation.

Furthermore, we find that in three of our four datasets, the use of evidential markers significantly improves performance. In fact, some of the best performing templates include evidential markers with a source. This is also consistent with recent work showing how the grounding of prompts increases model accuracy in generation (Weller et al., 2023). The top ten performing prompts for each dataset are listed in Tables 7, 8, 9, and 10.

The results across the other linguistic features are mixed. Across the four datasets, there is not a consistent improvement from the use of plausibility shields, sources, or personal pronouns (See Appendix).

For generalizability, we test five additional models (GPT3 Ada, Babbage, Curie, text-davinci-003, GPT4 (32k)) on the TriviaQA dataset (n=200) with our fifty templates and find our results reproduce. Across almost all models, boosters and factive verbs result in significant decreases to model performance and meanwhile evidentials markers significantly improve performance (Table 1).

## 4.2 Expressions of Uncertainty Compared to the Standard Prompting Method

Lastly, we find that the use of expressions of uncertainty might actually lead to better performance than the standard prompting method (i.e., just simply using "Q: <question> A:"). In TriviaQA the template *"Online says it's..."* achieves an accuracy of 66% compared to 63% achieved by the standard method. In Natural Questions, there are seven templates that outperform the standard method, six of which are expressions of uncertainty. Using our results with six models across all four datasets, we aggregated the results by template and identified six templates which perform significantly better ($t$-test; $p$-value < 0.05) than the standard method (Appendix Table 11) These promising results suggest that including uncertainty may not only help human decision makers, it may also improve the absolute accuracy of the model.

## 5 Why Does Certainty Hurt?

What explains our surprising finding that templates with weakeners outperform templates with strengtheners? Here, we discuss and evaluate hypotheses and potential confounders.

### 5.1 Certainty Affects Performance Independent of Perplexity

Recent work from Gonen et al. (2022) discusses how perplexity, as a proxy for frequency, explains variation in quality of generations. Phrases with high perplexity result in a significant drop in performance when used as prompts in language modeling. We test if perplexity could be a confounding variable in our experiments but find that the perplexity of our templates is in fact **not** correlated with the accuracy of the prompts (Pearson's $\rho$ = -0.03) (See Appendix). These results validate that the variations from prompts are not caused by trivial factors such as text length or the frequency of the expression in training data.

### 5.2 A Redistribution of Probability Mass When Prompted with Weakeners

Could it be that weakeners are changing the underlying probability distribution of the potential answers? If weakeners change the answer distribution, we might expect weakeners to induce an increase in the probability-on-gold (which is defined as the sum of the probabilities placed on all of the answer aliases). Focusing on GPT3 (davinci), we calculate the average probability-on-gold among all correctly predicted answers and find this is not the case. In fact, the probability-on-gold from templates with weakeners is slightly lower than the probability-on-gold from templates with strengtheners. This is true across three of our four datasets NaturalQA (42% vs 45%), JeopardyQA (47% vs 51%), and TriviaQA (53% vs 55%).

| Dataset | weakeners | strengtheners |
|---------|-----------|---------------|
| TriviaQA | **2.980** ± 0.01 | 2.917 ± 0.01 |
| CountryQA | **3.078** ± 0.02 | 2.875 ± 0.03 |
| Jeopardy | **3.170** ± 0.01 | 3.089 ± 0.01 |
| NaturalQA | **3.167** ± 0.01 | 3.106 ± 0.01 |

Table 2: Average entropy of the probability distribution of alternative tokens among weakeners and strengtheners. Across all four datasets, entropy is higher among weakeners, an indication the model places probability more evenly across the alternative answers. 95% CI calculated using standard error.

Furthermore, we find that weakeners led to a flattening of the distribution of probability mass across answers, compared to strengtheners. We look at the entropy of the probability distribution of top tokens not counting the top prediction; essentially the uncertainty among all but the top candidate. This entropy is significantly higher among weakeners than strengtheners (Table 2). Our finding suggests that the increase in accuracy of weakeners is not due to an increase in answer confidence, but rather when a weakener is used, the model responds by placing probability more evenly across each of the remaining possible options.

### 5.3 Certainty Used in Questions Instead of Answers

Why is it that expressions of *certainty* lead to lowered performance? We look for potential explanations by examining expressions of uncertainty in language model pretraining data. We queried for expressions of uncertainty like *"I'm certain it's"* or *"I'm sure it's"* in The Pile (Gao et al., 2020), a popular pretaining dataset, and in a qualitative exploratory analysis, found that certainty was often being used in questions rather than answers.

To quantitatively test this hypothesis, we measure the volume of expressions of uncertainty in the Stack Exchange section of the Pile with a set of uncertainty expressions. To our surprise, expressions of **certainty**, *"I'm sure"* and *"It must be"*, occur less than **half** as often in answers (104 instances per million words) as in questions (280 instances per million words). Conversely, expressions of **uncertainty**, *"It could be"* and *"Maybe it's"*, occur about twice as often in answers (436 instances per million words) as in questions (222 instances per million words). In other words, those asking for information are using expressions of certainty while those with the answers (or attempting to answer the question) are using expressions of uncertainty (details and statistics in Appendix).

A number of factors could be contributing to this phenomenon, including the use of uncertainty to express politeness (Brown and Levinson, 1987) or the use of certainty to rule out options, or the use of certainty to establish group membership (Hyland, 1998). To give an estimate of the prevalence of these acts, we perform qualitative open coding on 100 samples of two expressions. In question posts, the high certainty expression *"I'm sure"*, is used 34 times to establish group membership/admit ig-

norance (e.g., *"I'm sure it's a simple error"*) and used 22 times to isolate an issue (e.g., *"I'm sure I am catching it right"*). In answer posts, the low certainty expression *"I think"*, is used 25 times to politely give instructions and corrections (e.g., *"I think this should work for you:..."* and *"I think you meant to write..."*). These instances reveal the ways in which humans may be using expressions of certainty that go beyond expressing epistemic certainty. Our analysis of pretraining data suggests that language models might be mimicking this behavior and responding to prompts with epistemic markers in this surprising but explainable way.

## 6 The Impact of the Degree of Uncertainty on Performance

Results from Section 3.2 illustrate that GPT3 is highly sensitive to uncertainty in prompts, and certainty seems to lead to diminished accuracy. Here, we extend these results to study uncertainty at a more fine-grained level, allowing us to ask if the *degree of uncertainty* could play a role in the accuracy of model generation.

**Introducing Numerical Values**  Here, we introduce numerical values into our verbal expressions of uncertainty. The setup of our task changes from "What is the capital of Belarus? *I'm sure it's...*" to "What is the capital of Belarus? *I'm 90% sure it's...*". We use a set of seven expressions covering a range of numerical uncertainty expressions, including those that use personal pronouns to weaken uncertainty (*"I'm 90% certain..."*) and those which indicate uncertainty probabilistically but without mentioning the self (*"70% chance it's..."*). We also downsample our test set to 50 questions per dataset and evaluate each template at 0%, 10%, 30%, 50%, 70%, 90% and 100% intervals.

### 100% Certainty is not 100% Accurate

Our findings for numerical uncertainties extend our earlier analysis by enabling us to obtain more fine-grained results on whether a model is linguistically calibrated, and how much certainty causes performance degradation.

First, we find that numerical uncertainties in the prompt are not well calibrated with the accuracy of the answers generated.[7] E.g., the prompt *"I'm 90% sure it's..."* in TriviaQA only produces the correct

---

[7] Note this is slightly different from calibration in prior work which calibrates between the model's confidence and accuracy.

answer 57% of the time (Figure 4). Formally, we evaluate the expected calibration error (ECE) and find poor values ranging from 0.50 to 0.30, 0 being the best.

Second, consistent with our findings from Section 4, we find that certainty does hurt model performance. However, we find this is true only at the extremes. We see performance on models peaks usually between 70% and 90% in numerical values but drops in accuracy when using 100% in the prompt. Across all four datasets, with seven templates each, 21 out of the 28 templates which use 100% numerical values had a lower probability-on-gold than templates which used 90% numerical values (See Appendix). Additionally, at the other extreme, when 0% is used in templates, there is also a drastic drop to accuracy.

## 7 Why Does 100% Certainty Hurt?

We hypothesize this drop in accuracy might be the result of the use of hyperbolic or exaggerated language in the training set, in which numbers are used non-literally. When someone says *"I'm 100% certain there was pie left"*, they don't mean they are 100% certain is there pie — but rather are emphasizing their strong belief there was pie. Again, we turn to the Pile and qualitatively analyze over 500 examples of use cases of "100%". We remove other uses of "100%" such as in code (e.g., *"width:100%"*) and sample 50 instances from both question and answer posts. We find, surprisingly, that 100% is in fact often used to express **uncertainty**.

Of our 100 samples, 44 instances are used with negation (e.g., *"never be 100% accurate"* or *"is not always 100% reliable"*) and half are expressions of uncertainty (e.g., *"I'm not 100% sure"* or *"I don't 100% follow"*). In questions, the rate of expressions of uncertainty was nearly double that of answers (14 vs 8). In contrast, only 3 instances in our sample indicate 100% certainty (e.g., *"I'm 100% sure that"*). We hypothesize that both the use of negation with "100%" and the general lack of use of "100%" with expressions of certainty contribute to the lowered performance of these prompts.

Another confounder in how model's interpret numerical values could be the distribution of numerical frequencies in typical model training data. Querying the Pile, we find that there are drastic imbalances in the use of percentages in training datasets. There are significant spikes in frequency at the upper extremes (50%, 95% and 100%) (Fig-

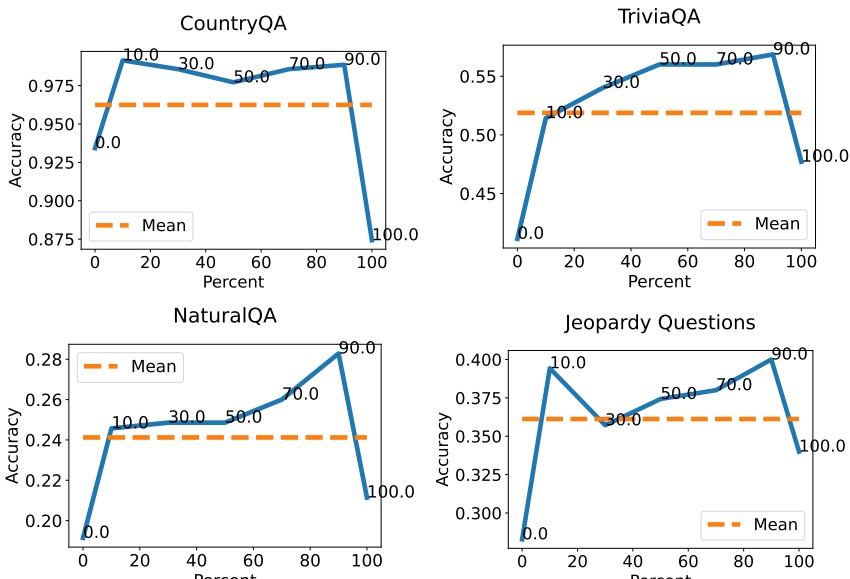

Figure 4: The X-axis indicates the percentage that was injected into the verbal uncertainty. The Y-axis indicates the accuracy across numerical uncertainties. Note the consistent drop in accuracy between 90% and 100% uncertainty and the increase in accuracy between 0% and 10% uncertainty.

ure 5). This might be happening as some values are more common to express (e.g., *"100% organic"*) or found more often in code. Humans might also naturally exaggerate values or use colloquial terms to describe confidence. The presence of scientific language like "95% confidence interval" could be another possible source of imbalance. Although spoken natural language also includes rounded percentages, the use of only textual data might be further exacerbating this bias.

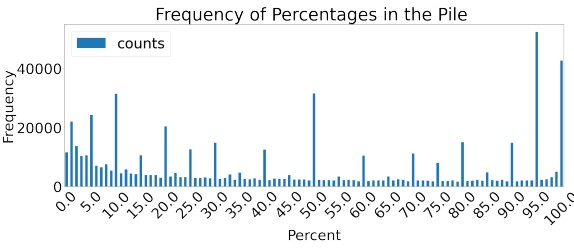

Figure 5: Visualization of the Frequency of percentages found in the pile. Note the peaks at the extremes, (0, 50, 10), and peaks at every 10 and 5 intervals from the first million samples queried from the Pile dataset using the HuggingFace API.

## 8 Related Work

While scholars have studied model uncertainty, prior work has focused on more accurately extracting model confidence (Kuhn et al., 2023; Sun et al., 2022; Gleave and Irving, 2022), measuring (Kwiatkowski et al., 2019b; Radford et al., 2019; Liang et al., 2023) and improving model calibration

(between model confidence and accuracy) (Jiang et al., 2021; Desai and Durrett, 2020; Jagannatha and Yu, 2020; Kamath et al., 2020; Kong et al., 2020). However, the community has found mixed results on the calibration of neural model (Minderer et al., 2021; Carrell et al., 2022); for example, Desai and Durrett (2020) shows that pretrained transformers are relatively well-calibrated meanwhile Wang et al. (2020) found severe miscalibration in neural machine translation. Another line of work also explore the trade-off between model performance and calibration (Stengel-Eskin and Van Durme, 2023).

Closest to our work, Mielke et al. (2022) propose solutions to reducing model overconfidence through linguistic calibration, Kadavath et al. (2022) experiment with models' ability emit self-confidence after finding that models are relatively well-calibrated, and Lin et al. (2022) teach models to be linguistically calibrated when answering math questions.

In addition, semanticists and computational linguists have long studied speaker commitment factors such as factivity (Karttunen, 1971; Degen and Tonhauser, 2022) and projection (Simons et al., 2010), and more recent work include corpora like the CommitmentBank (De Marneffe et al., 2019) which offers naturally occurring examples, as well as new experimental paradigms to investigate speaker commitment (Degen et al., 2019). A wide variety of scholars have examined computa-

tional issues in factuality, veridicality, and commitment (Saurí and Pustejovsky, 2009; de Marneffe et al., 2012; Stanovsky et al., 2017; Rudinger et al., 2018; Jiang and de Marneffe, 2021, inter alia) as well as bias (Pryzant et al., 2020; Patel and Pavlick, 2021) and specific devices like hedges (Prokofieva and Hirschberg, 2014; Raphalen et al., 2022), and modality (Pyatkin et al., 2021).

We see our work as a bridge between these two areas of speaker commitment and natural language generation, by telling us how models interpret speaker commitment through expressions of certainty and uncertainty.

## 9 Discussion and Conclusion

Here, we discuss a number of recommendations and future work for the greater NLP community.

**Can Models Generate Expressions of Uncertainty?** Having studied the impact of uncertainty as part of the prompt, a natural follow-up question is: how does model performance change when models learn to emit their own expressions of uncertainty? In preliminary experiments, we find it is challenging to calibrate models to generate epistemic markers. The same methods which work for numerical calibration do not transfer well to verbal calibration (Lin et al., 2022). However, we find some evidence that expressions of uncertainty are slightly better calibrated when compared with expressions of certainty. See A.1.

**Navigating Idiomatic Language** Humans use language that contains expressions of certainty when they are, in fact, *not* certain, and models appear to be mimicking this behavior. However, our QA models are still unable to recognize the meaning of those markers in our QA setup, raising concerns for how they would respond in downstream applications. Future work must seek to enable LMs to accurately interpret when expressions are meant idiomatically versus literally.

**Verified Attribution** We encourage the community to explore how to integrate attributions of information in a verified manner. Some of the best-performing templates from Section 4 include phrases like *"Wikipedia says..."*, however these were falsely injected attributions. As we move towards generating expressions of uncertainty, it is critical for researchers to be cautious when generating attributions at the risk of providing downstream users with a false, yet highly believable attribution.

In this work, we analyzed how epistemic markers impact model behavior. We find a drop in accuracy when naturalistic expressions of certainty (i.e., strengtheners and factive verbs) are present and trace this effect back to how expressions of certainty and uncertainty are used in pretraining datasets. As the community expands the capabilities of LMs to general language that is more natural, it is critical we prepare for the opportunity and harms that may arise from naturalistic expressions of uncertainty and certainty.

## Limitations

As with many language model work, some key limitations of our work include scaling to other models, increasing the variety of datasets, and experimenting with multi-shot prompting. The work and results we presented are robustly tested but if given additional resources (both in compute and time), the scalability of these results would be of value to the greater community.

Multi-shot prompting would dramatically increase the exploration space of epistemic markers, but this is a highly realistic scenario that should be explored in future work. Similarly, long-form and dialogue generation are both beyond the scope of this project but would further build our understanding how models interpret expressions of uncertainty.

Expressions of uncertainty also vary significantly across cultures and contexts and our study is limited by only studying how hedges and boosters exist in the English language. Syntatic, idomatic, and pragmtic differences in hedges could be interesting to study in follow-up work.

## Ethics Statement

Our work complies with ACL's ethical standard. We hope our work inspires future researchers to continue to investigate epistemic markers in language modeling. However, we have two critical ethical considerations when it comes future work regarding *generating* expressions of uncertainty.

First, we are concerned about the use of unverified attributions in language generation as the use of evidentials could appear to be very convincing to downstream users meanwhile providing little guarantee to the accuracy of the statements. Second, when it comes to generating epistemic markers, teaching models to only emit expressions of uncertainty when they are unsure,

rather than when they are sure, could be a safer design choice for human-computer interactions. Prior work has shown cases where AI-assisted decision-making performed worse than human decision-making alone, suggesting an over-reliance on AI, even when systems are wrong (Jacobs et al., 2021; Bussone et al., 2015). Teaching models to emit expressions of certainty could further exacerbate these challenges.

## Acknowledgements

Thank you so much to Tolúlo̩pé̩ Ò̩gúnrè̩mí, Yiwei Luo, Myra Cheng, Rishi Bommasani, and Omar Shaikh for their helpful feedback and advice! Thank you to Center for Research on Foundation Models (CRFM) for their generous support, and to the National Science Foundation for funding via award IIS-2128145. Tatsunori Hashimoto and Dan Jurafsky were supported by a gift from Open Philanthropy. Kaitlyn Zhou is supported by the Stanford Graduate Fellowship.

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

# A Appendix

## Additional Details

### A.1 When LMs Emit Their Own Uncertainty

Here, we study how model performance changes based on in-context learning examples. Specifically, we follow Lin et al. (2022)'s method in few-shot learning with 50 samples which has been shown to be nearly as effective as fine-tuning on datasets that are magnitudes larger. To ensure that our in-context learning dataset covers a range of confidence levels, our dataset contains 48 samples whose probability-on-gold is uniformly distributed (in buckets of 10) between 0 and 100.

#### A.1.1 Experiment Details

To study how LMs respond when emitting their own uncertainty, we follow Lin et al. (2022)'s setup but modify it for the strengtheners and weakeners, which are inherently non-numerical (Figure 1). In our setting, instead of teaching a model to output a percentage confidence, we teach it to output a strengthener when the confidence is above a threshold and nothing otherwise.[8] Conversely, when we study weakeners, we teach it to output a weakener when the probability is below a threshold and nothing otherwise.

As an example, consider the question "What is the capital of France". We record the LM's probability over Paris (the probability-on-gold) and append "*I'm sure*" to the in-context example if the model's confidence was above 0.5. We repeat this for all the in-context examples to obtain our in-context learning training set.

#### A.1.2 Prompting Perturbations

Recent work has shown the drastic differences that appear based on simple changes to prompting setup (Suzgun et al., 2022; Lu et al., 2022). We design our in-context learning samples with various perturbations to ensure robustness in our results.

We select high performing weakeners and strengtheners from Section 4 and experiment with appending expressions of uncertainty after the answer (e.g., "Paris. I think") (Table 12).[9] We then use three different sample orderings to perturb our learning samples: ascending and descending order

of probability-on-gold and random ordering.[10] Finally, we experiment with a variety of thresholds (0.3, 0.5, 0.7, 0.9) for determining when expressions of (un)certainty should be inserted into the example. These perturbations are done on a small scale to help identify the best hyper-parameters (threshold, placement, and ordering) to use.

We find that varying the threshold across does not drastically change accuracy (with all methods attaining ∼ 83% in accuracy), although the threshold does significantly impact the balance of the training datasets. Similarly, we find limited differences in the ordering of the samples. With these results, we choose a threshold of 0.5 (creating a balanced dataset) and random ordering (simplest setting) as our hyper-parameters for our remaining scenarios and analysis, which we test on 100 TriviaQA questions.

#### A.1.3 Results

**Gains in Model Calibration When Learning Uncertainty** Overall, GPT3 has a limited ability to learn naturalistic expressions of uncertainty in a calibrated manner. We measured calibration based on whether models successfully emit the expressions of uncertainty when the probability of the top token above or below our training threshold. In our setup, when learning to emit certainty, answers with probability-on-gold of greater than 0.5 had a strengthener and answers with less than 0.5 had nothing. Therefore, in its generation when the probability on the top token is greater than 0.5, we'd expect the model to also generate a strengthener and vice versa for weakeners. We measure whether the model successfully generates strengtheners and weakeners through the F1 score, and find that the template with the highest macro-F1 score for uncertainty templates to be 0.56 compared to 0.53 for certainty templates.[11] This is close to the random guessing baseline on this test set which results in an F1 score of 0.45.

To illustrate the difference in calibration between uncertainty and certainty, we can look at the average accuracy when a model emits an expression or not. When learning to express weakeners, the generation of a weakener results in an accuracy of 74% but this increases to 83% when the model doesn't

---

[8]We choose to emit nothing rather than emit an expression of uncertainty, as we wish to isolate the effect of each linguistic expression of uncertainty.

[9]Here, we exclude expressions with attribution shields for concerns of false attribution, more on this in the discussion.

[10]In the ascending and descending orders, all the samples in the beginning or all the samples at the end will include expressions of (un)certainty.

[11]Average F1 scores being .52 average for uncertainty and .49 average of certainty

| Entropy | Emitting | Not Emitting |
|---|---|---|
| Uncertainty | **0.699*** | 0.522 |
| Certainty | 0.461 | **0.617*** |
| Control | N/A | 0.541 |

| Accuracy | Emitting | Not Emitting |
|---|---|---|
| Uncertainty | 0.738 | **0.829*** |
| Certainty | 0.799 | 0.789 |
| Control | N/A | 0.78 |

Table 3: Entropy is higher when uncertainty is being expressed and also higher when certainty is not expressed. Accuracy is also higher when uncertainty is not expressed but accuracy is not significant higher when certainty is expressed (an indication or poor calibration). *Significantly higher value calculated using two-sample t-test, $p < 0.05$.

generate a weakener. This is the intended behavior, with the model hedging answers it is more likely to get incorrect. However, when teaching the model to emit strengtheners, the generation of strengtheners does not lead to a significant increase in accuracy (79% with or without an emission of strengtheners). This means that when the model emits certainty, the answer is not more likely to be correct, creating a concerning issue for linguistic calibration (Table 3).

**Modeling Changes in Entropy**  Despite low model calibration when emitting expressions of certainty and uncertainty, we find that the underlying entropy of the probability distribution of the generated answer is well-calibrated to the expressions of uncertainty. Analyzing the top five predictions for each token, we find that when teaching models weakeners, the entropy of the distribution of potential generations is higher when a weakener is emitted and lower when it is not. The inverse is true when teaching models strengtheners, entropy is lower when strengtheners are emitted and higher when it is not. Although the calibration scores are not strong for either uncertainty or certainty, we see promising behaviors in the entropy of the model's top generations. When emitting weakeners, the model places more consideration on alternative answers and less when emitting strengtheners.

**Sensitivity to Placement of Template**  Finally, we test how model performance differs based on the placement of the templates. The simple design

difference of probing for the answer before (e.g., "I think it's Paris.") or after (e.g., "Paris. I think.") an expression of uncertainty can have a significant difference in performance. In our tables, we refer to these places as prefixes (before) and suffixes (after). We find that when appending expressions of uncertainty as a prefix, the generation is significantly worse for accuracy (63% vs 80%). This is also correlated with probability-on-gold being lower in prefixed templates (40% vs 67%). An explanation for this might be that the probability of generating the correct answer will be lower if generated after a phrase like "I think it's…" rather than just generated immediately after the question. Our work suggests that ordering effects may be important when addressing accuracy-calibration trade-offs in LMs and that there are accuracy gains when prompting the model to respond with answers as soon as possible.

| Template | Certainty | Prob | Top 1 |
|---|---|---|---|
| Prefix | Uncertain | 0.388 | 0.592 |
|  | Certain | 0.407 | 0.674 |
| Suffix | Uncertain | **0.674** | **0.800** |
|  | Certain | 0.673 | 0.792 |
| Control | N/A | 0.673 | 0.780 |

Table 4: Average probability on generated token and top 1 accuracy across prefix and suffix templates.

## A.2  Perplexity vs Accuracy

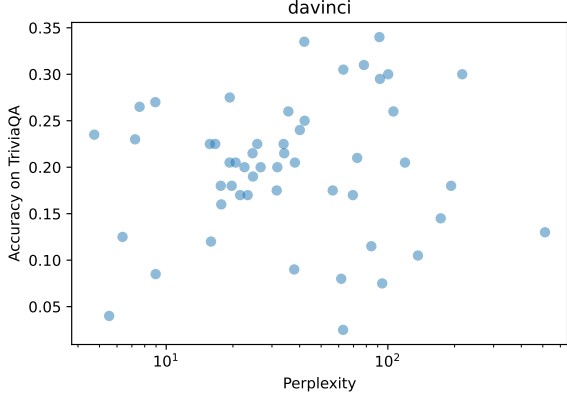

Figure 6: Correlation between the perplexity (GPT3 davinci) and the accuracy of an expression of uncertainty on questions from TriviaQA (Pearson's $\rho$ = -0.03).

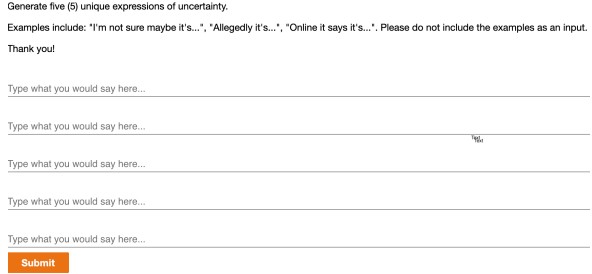

Figure 7: Screenshot of the Crowdsourced Example

## A.3 Performance of Additional Linguistic Features

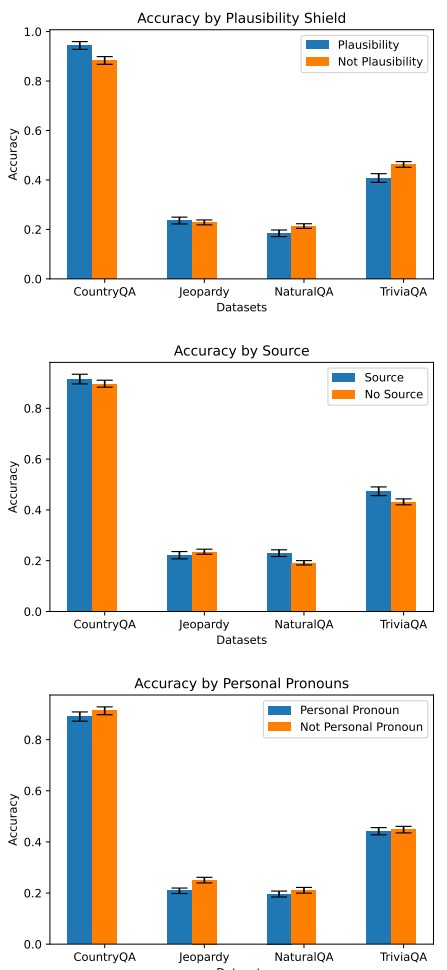

Figure 8: The use of plausibility shields, sources, and personal pronouns are mixed, without significant consistent improvements or drops in accuracy. 95% CI calculated using bootstrap resampling.

## A.4 Additional Tables and Results

| Expressions | uncertainty | # instances | # per thousand posts | # per million words | # instances | # per thousand posts | # per million words |
| --- | --- | --- | --- | --- | --- | --- | --- |
| | | | Questions | | | Answers | |
| i think | hedge | 1,106,442 | 37.5 | 162.2 | 1,536,543 | 52.0 | 302.7 |
| it could be | hedge | 84,239 | 2.9 | 12.3 | 143,670 | 4.1 | 28.3 |
| it might be | hedge | 70,606 | 2.4 | 10.3 | 170,803 | 4.9 | 33.6 |
| maybe it's | hedge | 21,803 | 0.7 | 3.2 | 17,233 | 0.5 | 3.4 |
| it should be | hedge | 233,686 | 7.9 | 34.3 | 346,290 | 10.0 | 68.2 |
| **Total** | | **1,516,776** | **51.4** | **222.3** | **2,214,539** | **63.9** | **436.2** |
| i know | booster | 1,672,756 | 56.6 | 245.2 | 350,241 | 10.1 | 69.0 |
| i'm certain | booster | 5,975 | 0.2 | 0.9 | 2,758 | 0.1 | 0.5 |
| i am certain | booster | 4,638 | 0.1 | 0.7 | 1,607 | 0.0 | 0.3 |
| i'm sure | booster | 119,224 | 4.0 | 17.5 | 76,009 | 2.2 | 15.0 |
| i am sure | booster | 52,089 | 1.8 | 7.6 | 22,983 | 0.7 | 4.5 |
| it must be | booster | 52,976 | 1.8 | 7.8 | 72,724 | 2.1 | 14.3 |
| evidently it's | booster | 33 | 0.0 | 0.0 | 52 | 0.0 | 0.0 |
| **Total** | | **1,907,691** | **64.58** | **279.6** | **526,374** | **15.2** | **103.7** |

Table 5: Counts of expressions of certainty and uncertainty in the Stack Exchange section of The Pile.

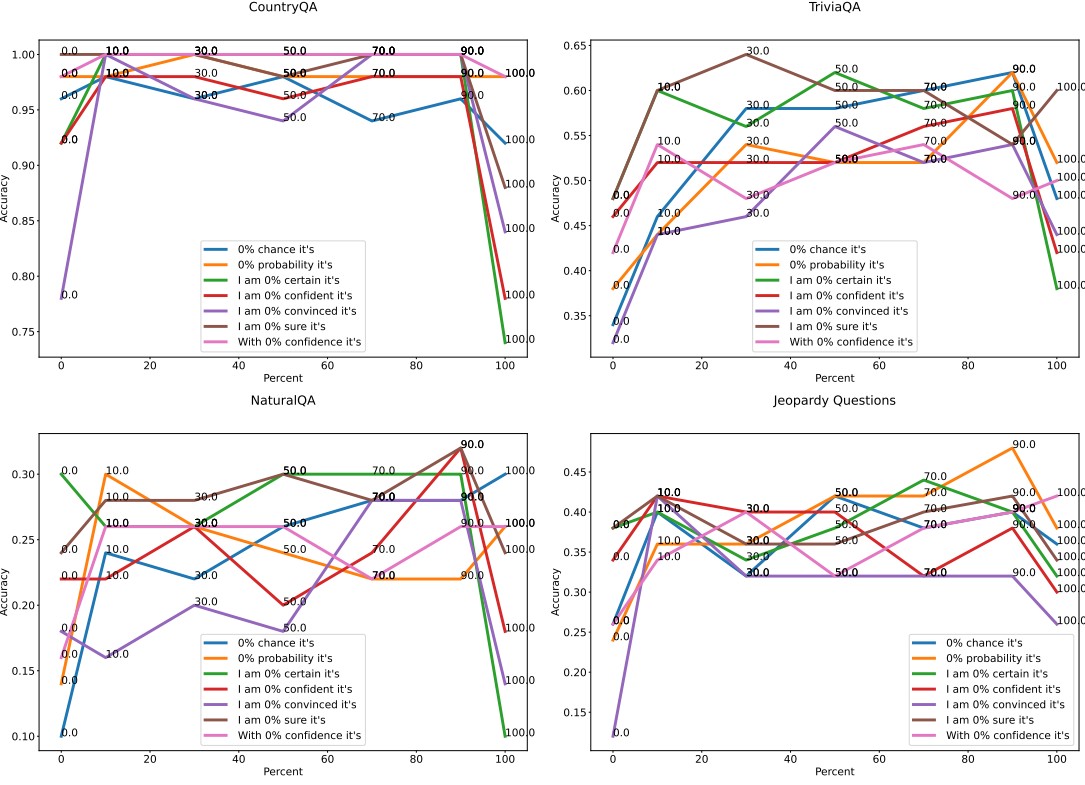

Figure 9: Variation in probability-on-gold across numerical uncertainties. Note the consistent drop in accuracy between 90% and 100% uncertainty and the increase in accuracy between 0% and 10% uncertainty.

| Template | Strengtheners | Shield | Evidential Marker | Factive Verb | Source | 1P |
|---|---|---|---|---|---|---|
| Apparently it's | Weakener | None | Evidential | Not Factive | No Source | No |
| Rumor says it it's | Weakener | None | Evidential | Not Factive | No Source | No |
| Allegedly it's | Weakener | None | Evidential | Not Factive | No Source | No |
| I was told it's | Weakener | None | Evidential | Not Factive | No Source | Yes |
| I've heard it's | Weakener | None | Evidential | Not Factive | No Source | Yes |
| They told me it's | Weakener | None | Evidential | Not Factive | No Source | Yes |
| Wikipedia suggests it's | Weakener | None | Evidential | Not Factive | Source | No |
| Online says it's | Weakener | None | Evidential | Not Factive | Source | No |
| The internet says it's | Weakener | None | Evidential | Not Factive | Source | No |
| Wikipedia claims it's | Weakener | None | Evidential | Not Factive | Source | No |
| Wikipedia says it's | Weakener | None | Evidential | Not Factive | Source | No |
| I read on the internet it's | Weakener | None | Evidential | Not Factive | Source | Yes |
| I read on Wikipedia it's | Weakener | None | Evidential | Not Factive | Source | Yes |
| I read online it's | Weakener | None | Evidential | Not Factive | Source | Yes |
| Presumably it's | Weakener | None | Not Evidential | Not Factive | No Source | No |
| To the best of my knowledge it's | Weakener | Plausibility | Evidential | Not Factive | No Source | Yes |
| As far as I'm aware it's | Weakener | Plausibility | Evidential | Not Factive | No Source | Yes |
| I vaguely remember it's | Weakener | Plausibility | Evidential | Not Factive | No Source | Yes |
| It could be | Weakener | Plausibility | Not Evidential | Not Factive | No Source | No |
| Considering all the options it's | Weakener | Plausibility | Not Evidential | Not Factive | No Source | No |
| It probably is | Weakener | Plausibility | Not Evidential | Not Factive | No Source | No |
| Maybe it's | Weakener | Plausibility | Not Evidential | Not Factive | No Source | No |
| Perhaps it's | Weakener | Plausibility | Not Evidential | Not Factive | No Source | No |
| It should be | Weakener | Plausibility | Not Evidential | Not Factive | No Source | No |
| I don't know maybe it's | Weakener | Plausibility | Not Evidential | Not Factive | No Source | Yes |
| I suppose it's | Weakener | Plausibility | Not Evidential | Not Factive | No Source | Yes |
| I would need to double check but maybe it's | Weakener | Plausibility | Not Evidential | Not Factive | No Source | Yes |
| I wouldn't put money on it but maybe it's | Weakener | Plausibility | Not Evidential | Not Factive | No Source | Yes |
| I'm not an expert but maybe it's | Weakener | Plausibility | Not Evidential | Not Factive | No Source | Yes |
| I think it's | Weakener | Plausibility | Not Evidential | Not Factive | No Source | Yes |
| I feel like it should be | Weakener | Plausibility | Not Evidential | Not Factive | No Source | Yes |
| It is known that it's | Strengthener | None | Evidential | Factive | No Source | No |
| The most recent evidence shows it's | Strengthener | None | Evidential | Factive | Source | No |
| The rules state it's | Strengthener | None | Evidential | Factive | Source | No |
| Two recent studies demonstrate it's | Strengthener | None | Evidential | Factive | Source | No |
| Wikipedia acknowledges it's | Strengthener | None | Evidential | Factive | Source | No |
| Wikipedia confirms it's | Strengthener | None | Evidential | Factive | Source | No |
| Our lab has shown it's | Strengthener | None | Evidential | Factive | Source | Yes |
| Evidently it's | Strengthener | None | Evidential | Not Factive | No Source | No |
| According to the latest research it's | Strengthener | None | Evidential | Not Factive | Source | No |
| We can see in the textbook that it's | Strengthener | None | Evidential | Not Factive | Source | Yes |
| It must be | Strengthener | None | Not Evidential | Factive | No Source | No |
| We realize it's | Strengthener | None | Not Evidential | Factive | No Source | Yes |
| We understand it's | Strengthener | None | Not Evidential | Factive | No Source | Yes |
| We know it's | Strengthener | None | Not Evidential | Factive | No Source | Yes |
| Undoubtedly it's | Strengthener | None | Not Evidential | Not Factive | No Source | No |
| With 100% confidence it's | Strengthener | None | Not Evidential | Not Factive | No Source | No |
| I'm certain it's | Strengthener | None | Not Evidential | Not Factive | No Source | Yes |
| I am 100% sure it's | Strengthener | None | Not Evidential | Not Factive | No Source | Yes |
| It's | None | None | Not Evidential | Not Factive | No Source | No |

Table 6: Full list of expressions of uncertainty coded for six linguistic features. *Claims is a neg-factive but in our schema, will just be considered not a factive verb. (Saurí and Pustejovsky, 2009)

| | Template | Type | Top 1 Accuracy |
|---|---|---|---|
| 0 | Online says it's | Weakener, Evidential,Source | 0.660 |
| 1 | Standard Method | - | 0.625 |
| 2 | Wikipedia confirms it's | Strengthener, Evidential, Factive, Source | 0.600 |
| 3 | Wikipedia suggests it's | Weakener, Evidential, Source | 0.595 |
| 4 | The internet says it's | Weakener, Evidential, Source | 0.585 |
| 5 | Wikipedia claims it's | Weakener, Evidential, Source | 0.575 |
| 6 | Wikipedia says it's | Weakener, Evidential, Source | 0.575 |
| 7 | We can see in the textbook that it's | Strengthener, Evidential, Source, 1P | 0.565 |
| 8 | I would need to double check but maybe it's | Weakener, Plausibility, 1P | 0.555 |
| 9 | Rumor says it it's | Weakener, Evidential | 0.550 |

Table 7: Top 10 Templates For TriviaQA for GPT3 - Davinci

| | Template | Type | Top 1 Accuracy |
|---|---|---|---|
| 0 | Standard Method | - | 1.0 |
| 1 | I read on Wikipedia it's | Weakener, Evidential, Source, 1P | 1.0 |
| 2 | It's | - | 1.0 |
| 3 | It should be | Weakener, Plausibility | 1.0 |
| 4 | Allegedly it's | Weakener, Evidential, | 1.0 |
| 5 | I'm not an expert but maybe it's | Weakener, Plausibility, 1P | 1.0 |
| 6 | I wouldn't put money on it but maybe it's | Weakener, Plausibility, 1P | 1.0 |
| 7 | Presumably it's | Weakener | 1.0 |
| 8 | I read online it's | Weakener, Evidential, Source, 1P | 1.0 |
| 9 | I read on the internet it's | Weakener, Evidential, Source, 1P | 1.0 |

Table 8: Top 10 Templates For CountryQA for GPT3 - Davinci

| | Template | Type | Top 1 Accuracy |
|---|---|---|---|
| 0 | Standard Method | - | 0.450 |
| 1 | It must be | Strengthener, Factive | 0.390 |
| 2 | It's | - | 0.380 |
| 3 | It could be | Weakener, Plausibility | 0.370 |
| 4 | The internet says it's | Weakener, Evidential, Source | 0.370 |
| 5 | Online says it's | Weakener, Evidential, Source | 0.360 |
| 6 | With 100% confidence it's | Strengthener | 0.350 |
| 7 | Undoubtedly it's | Strengthener | 0.345 |
| 8 | Wikipedia says it's | Weakener, Evidential, Source | 0.345 |
| 9 | Wikipedia confirms it's | Strengthener, Evidential, Factive, Source | 0.325 |

Table 9: Top 10 Templates for Jeopardy for GPT3 - Davinci

| | Template | Type | Top 1 Accuracy |
|---|---|---|---|
| 0 | Wikipedia claims it's | Weakener, Evidential, Source | 0.340 |
| 1 | Wikipedia says it's | Weakener, Evidential, Source | 0.335 |
| 2 | Online says it's | Weakener, Evidential, Source | 0.310 |
| 3 | Wikipedia suggests it's | Weakener, Evidential, Source | 0.305 |
| 4 | The internet says it's | Weakener, Evidential, Source | 0.300 |
| 5 | Wikipedia confirms it's | Strengthener, Evidential, Factive, Source | 0.300 |
| 6 | I read on Wikipedia it's | Weakener, Evidential, Source, 1P | 0.295 |
| 7 | Presumably it's | Weakener | 0.275 |
| 8 | Standard Method | - | 0.275 |
| 9 | I think it's | Weakener, Plausibility, 1P | 0.270 |

Table 10: Top 10 Templates for NaturalQA for GPT3 - Davinci

| | Template | Type | Top 1 Accuracy |
|---|---|---|---|
| 0 | The internet says it's | Weakener, Evidential, Source | 0.416 |
| 1 | Wikipedia says it's | Weakener, Evidential, Source | 0.408 |
| 2 | Online says it's | Weakener, Evidential, Source | 0.405 |
| 3 | Wikipedia suggests it's | Weakener, Evidential, Source | 0.404 |
| 4 | Wikipedia claims it's | Weakener, Evidential, Source | 0.400 |
| 5 | Wikipedia confirms it's | Strengthener, Evidential, Factive, Source | 0.397 |
| 6 | I would need to double check but maybe it's | Weakener, Plausibility, 1P | 0.387 |
| 7 | I'm not an expert but maybe it's | Weakener, Plausibility, 1P | 0.387 |
| 8 | I am 100% sure it's | Strengthener, 1P | 0.385 |
| 9 | We can see in the textbook that it's | Strengthener, Evidential, Source, 1P | 0.382 |

Table 11: Top 10 Templates Across All GPT Models and All Datasets

| Expression | Suffix | Prefix |
|---|---|---|
| Certainty | Undoubtedly. | Undoubtedly it's |
| Certainty | With 100% confidence. | With 100% confidence it's |
| Certainty | We know it. | We know it's |
| Certainty | Evidently. | Evidently it's |
| Certainty | It must be. | It must be |
| Uncertainty | I think. | I think it's |
| Uncertainty | It could be. | It could be |
| Uncertainty | But I would need to double check. | I would need to double check but maybe it's |
| Uncertainty | I suppose. | I suppose it's |
| Uncertainty | But I wouldn't put money on it. | I wouldn't put money on it but maybe it's |

Table 12: List of Templates Used for Section A.1