# OpenReview forum: "Navigating the Grey Area: How Expressions of Uncertainty and Overconfidence Affect Language Models"
_EMNLP/2023/Conference — EMNLP 2023 Main_

### Official Review · Reviewer_SvKk · 2023-08-03

**Soundness:** 3

**Excitement:**

3: Ambivalent: It has merits (e.g., it reports state-of-the-art results, the idea is nice), but there are key weaknesses (e.g., it describes incremental work), and it can significantly benefit from another round of revision. However, I won't object to accepting it if my co-reviewers champion it.

**Paper Topic And Main Contributions:**

This paper investigates the impact of epistemic expressions, including certainty, uncertainty, and evidentiality, on language models. It achieves the goal through an innovative approach, employing zero-shot prompting and infusing verbal and numerical markers into trivia question prompts.
This is an interesting topic because it examines how specific linguistic issues would affect language models.


**Reasons To Accept:**

This paper explores the influence of epistemic expressions on language models using zero-shot prompting, an innovative approach.



**Reasons To Reject:**

However, the potential of generalization would be questionable given the limited range of epistemic expressions under examination.

**Reproducibility:**

4: Could mostly reproduce the results, but there may be some variation because of sample variance or minor variations in their interpretation of the protocol or method.

**Reviewer Confidence:**

2: Willing to defend my evaluation, but it is fairly likely that I missed some details, didn't understand some central points, or can't be sure about the novelty of the work.

---

> ### Author Rebuttal · Authors · 2023-08-28
>
> * Thank you for the review and agreeing this is an interesting topic and highlighting our innovative approach!
> ###### Range of Epistemic Markers:
> * We cover a range of epistemic markers as covered in the literature and cited in our paper. We wanted to cover a range of common linguistic features related to epistemic markers and had examples related to each broad linguistic category. We think there is a lot of potential for great follow up work in additional epistemic expressions.

---

### Official Review · Reviewer_K4oh · 2023-08-04

**Typos Grammar Style And Presentation Improvements:** 1.	Typo on line 395
**Soundness:** 4

**Excitement:**

4: Strong: This paper deepens the understanding of some phenomenon or lowers the barriers to an existing research direction.

**Missing References:**

N/A

**Paper Topic And Main Contributions:**

The paper studies how including expression of certainty (such as, “I’m 100% sure it’s”) and expressions of uncertainty (such as, “Maybe it’s”) in the model prompt in question answering setup affect the model’s accuracy. The authors collect a set of naturally occurring expressions of certainty/uncertainty from literature in linguistics and experiment with these expressions in zero-shot prompting setup. The analysis presented in the paper reveals that contrary to expectations, model performance drops on addition of expression of certainty to the prompt. The paper conducts further analysis to identify reason for this behavior and find that in the pre-training data, the expressions of uncertainty are frequently used in answers (possibly to portray politeness). Expressions of certainty appear more frequently in questions (possibly to rule out incorrect answers). They hypothesize this as a possible cause of the model’s unexpected behavior. The main takeaway of the paper is that LLMs might not be calibrated to expressions of uncertainty, possibly drawing questions on the ability of these models to generate expressions of uncertainty that aptly represents the model accuracy.

**Questions For The Authors:**

1.	The paper describes the results across multiple linguistic phenomenon: whether the expression is a strengthener (reflects certainty) or weakener (reflects uncertainty), whether the expression includes an evidential marker (for e.g., “Wikipedia suggests it’s”) and whether the expression is factive, that is, it presupposes the truth of its complement cause (e.g., “I realized”). Is the effect of expression of uncertainty mediated by the presence of evidential markers? From Tables 7-10, it seems that most of the top templates from the weakener class are also evidential in nature. There might be an interesting underlying phenomenon at play here, given that most of the evidential markers of the strengthener type are also factive. Maybe, the difference in the accuracy for strengthener and weakener class is explained away by the other properties itself?

**Reasons To Accept:**

1.	The findings in the paper are quite interesting and highlights a major flaw in LM behavior with respect to epistemic markers of uncertainty, that are contrary to expectations.
2.	Overall, the paper conducts very thoroughly analysis, not only to draw out the LM behavior, but also to investigate potential sources of this behavior, leading to useful insights into the problem.


**Reasons To Reject:**

1.	Some assumptions in the evaluation might be resulting in undesirable scoring. The paper mentions that “while calculating accuracy, we generate 10 tokens and if any of these tokens matches the answer, we’ll count that as a correct generation. This is done to not unfairly disadvantage templates that are prone to generating words prior to emitting the answer (e.g., ‘Allegedly, it’s said to be…’)”. My concern is that this disregards that LM might output something “Allegedly, it’s believed to be … but facts show that it’s”, or something to this extent, in which case the model would be awarded for either answers. It is also likely that when prompted with expressions of uncertainty, the model might generate multiple possible answers (e.g., “I am not sure, but it’s either … or …), again giving model higher points. This might also be a potential reason for higher accuracy when models are prompted with expressions of uncertainty than expressions of certainty. This evaluation needs further validation to ensure that no unintentional behavior is captured here.
2.	Minor issue, but the paper does not mention a lot of details about the experimental setting. Mainly, is the model prompted with each template, question pair only once. Are the trends consistent on multiple prompting?


**Reproducibility:**

3: Could reproduce the results with some difficulty. The settings of parameters are underspecified or subjectively determined; the training/evaluation data are not widely available.

**Reviewer Confidence:**

4: Quite sure. I tried to check the important points carefully. It's unlikely, though conceivable, that I missed something that should affect my ratings.

---

> ### Author Rebuttal · Authors · 2023-08-28
>
> * One line summary: We addressed the reviewer's main concern by qualitatively sampling responses and found that the potential undesirable behavior is very infrequent, ensuring the accuracy and robustness of our scoring.
> * Thank you for your positive review! Thank you for supporting this paper and pointing out that the paper highlights a major flaw in LM behavior. We also thank you and reviewer 1 for both pointing out the thorough analysis that was done to draw out the LM behavior as well as to investigate the potential sources. We hope that our methods, artifacts, and insights will equip the community on best steps forward to addressing this issue. We really appreciate the time you took to engage with our work and for helping us more rigorously present our findings and results.
> ###### Scoring process:
> * We manually inspected 1,000 answers to check for this behavior. We found only one instance of this occurring where the first token was “not” and the second token was the correct answer, indicating that this is a very rare behavior. Thank you for pointing this out and helping us be extra rigorous with our results.
> ###### Experimental details and consistency across prompts
> * For all models except GPT4, we set temperature to 1 and took the most probable token associated with each generation which should lead to mostly consistent results across multiple prompts. This is in contrast to just taking the generated output which would lead to different generations at each prompting. For GPT4, where the logprobs are currently not available, we set temperature to 0, also ensuring a (mostly) deterministic output.
> ###### Questions For The Authors
> * This is a great and keen observation! Given our current set of templates, we don't think we have the statistical power to fully answer how hedges and boosters would perform without the influence of evidentials and factives. We have a limited number of booster templates that are non-factive and non-evidential, making it difficult to fully explore this question. Based on some taxonomies, including the ones we cite (Prince et. al. 1982), evidentials are considered to be hedges as the speaker is not fully committing to the statement but rather attributing part of it to another source. It is interesting to think about hedges that don't contain evidentials and boosters that don't contain factives, but we think this exploration requires consideration of other potential confounders which are best suited for future work. One suggestive piece of evidence we can say from section 6 is that in the numerical uncertainty setting, which contained templates without factives and evidentials, we know that absolute certainty performs worse than hedged expressions.
> ###### Typos
> * This is very helpful and will take care of these issues for the final version.

---

### Official Review · Reviewer_JAsR · 2023-08-05

**Soundness:** 4

**Excitement:**

4: Strong: This paper deepens the understanding of some phenomenon or lowers the barriers to an existing research direction.

**Paper Topic And Main Contributions:**

This paper examines how LLMs respond to epistemic markers of certainty and uncertainty.  To this end, the authors develop a list of 50 different modifiers based on linguistic theory, which they insert into QA prompts. They explore two main hypotheses: 1. LLMs are robust to these modifiers. 2. LLMSs respond to the modifiers in that the certainty of the modifiers increases accuracy and vice versa. The experiments contradict both hypotheses. 1. LLMs (GPT-3 in the focus of this work) are not robust to the markers, as indicated by significant changes in accuracy. 2. LLMs respond to the markers in the opposite way as hypothesized.  The authors then analyze in depth why LLMs behave as encountered by looking at the correlation with prompt perplexity, whether the underlying probability distribution is altered by the markers, and the extent to which certainty and uncertainty are present in questions and answers included in QA pre-training datasets. Moreover, they add degrees of uncertainty in the form of percentages to specifically analyze the impact of the spectrum on LLM performance. This extended analysis provides valuable insights into the behavior of LLMs with respect to epistemic markers of certainty and uncertainty.

**Questions For The Authors:**

- A: Why did the authors refrain from using approximators in their work (introduced in Section 2)?
- B: Will you make the two new datasets available to the community? Will you report which 200 respectively 53 QA pairs you chose from the four datasets?

**Reasons To Accept:**

- Analyzing the behavior of LLMs with respect to specific linguistic phenomena is of great interest and value to the community in order to better understand the performance of these models and discover potential pitfalls for practical application. This work contributes to this by focusing on markers of certainty and uncertainty.
- The paper is well written and easy to follow. The motivation of the paper and included experiments is very clear.
- The contrasting findings of the study with respect to the two intuitive working hypotheses are intriguing.
- Particularly precious in this work is the detailed analysis of the why in Sections 5 to 7, which reveals, for example, unfavorable properties of the pre-training dataset that may provoke the unexpected effect.

**Reasons To Reject:**

- Section 2.1: The process of building the list of epistemic markers is very unclear. It should be explained in detail which input came from the literature, which came from the authors and with which substantiation. Furthermore, the description of crowd sourcing some of the markers should be motivated and included in the main body of the paper rather than in the appendix. [convincingly addressed in the rebuttal]
- The model setup should be reported in detail, such as which pre-trained models are exactly used (e.g., add a link to the model if they are from huggingface).
- The evaluation design was seriously restricted in that models other than GPT-3 were evaluated on only one of four datasets. Since the datasets are small, this limitation is not understandable and reduces the significance of the results. [convincingly addressed in the rebuttal]
- Section 4.2: The conclusiveness of the results is questionable, as they appear to be single observations. Therefore, they need to be confirmed, for example, with a regression analysis.

Minor issues:
- line 449-452: It is not clear why the test set was downsampled. Please explain your decision. [justification provided in the rebuttal]

**Reproducibility:**

3: Could reproduce the results with some difficulty. The settings of parameters are underspecified or subjectively determined; the training/evaluation data are not widely available.

**Reviewer Confidence:**

4: Quite sure. I tried to check the important points carefully. It's unlikely, though conceivable, that I missed something that should affect my ratings.

**Typos Grammar Style And Presentation Improvements:**

- Location of "Related Work": There is a very early reference to this in Section 2, so it seems much more logical to put the "Related Work" section up front. (i.e., Section 2 "Related Work", Section 3 "Expressions of Certainty and Uncertainty: Linguistic Background").
- Figure 2: Include all groups of markers.
- In several sections there is only one subsection. Usually one has none or several. You may think about modifying this accordingly.
- There is a repeated use of reference to “Appendix”. This is not very helpful as the appendix is extensive. Please include the exact section in these cases.
- The location of Figure 1 is a bit unfortunate as it is referenced pages after. However, I feel that the content of the figure fits well in the introduction, so the reference could be made accordingly earlier.
- Shift Table 2 into Section 5.2.
- line 466: “Figure 4”
- Section 7 title: 100% of what? (certainty?)
- In general, it would be much easier for the reader if the sections in the appendix were arranged chronologically as they are referenced in the main body.
- line 395: There is a content error here: “twice as often” should obviously be replaced by “half as often”.

---

> ### Author Rebuttal · Authors · 2023-08-28
>
> * One line summary: We conducted new experiments across all four datasets on six models, and our results reproduced nearly exactly, further confirming our findings.
> * We thank the reviewer and agree that analyzing the behavior of LLMs with respect to epistemic markers is of great value to the community and has significant practical implications moving forward. We’ve conducted a number of experiments following your comments which have helped us strengthen the rigor and soundness of our findings. Details below.
> ###### Process of Building Epistemic Markers
> * While there’s a diverse literature on epistemic markers, there is no single coherent taxonomy in the literature. Therefore, we engaged in a bottom-up process where we gathered a diverse set of templates guided by the linguistic literature and used the literature to develop our taxonomy and classify our templates. To gather the templates, the authors relied on linguistic literature like Prince et al. 1982, Amazon Mechanical Turk, and brainstorming among the authors. Next, we drew on additional linguistic literature to augment this list, adding dimensions such as factive verbs (Kiparsky and Kiparsky, 1970), evidentials (a broader category of attributions) (Aikhenvald, 2004, 2020), and authoritative sources. We then integrated this literature to develop a coherent taxonomy and classify each of our templates.
> * We aim to use the extra page in the final version of the paper to include these details. Thank you for highlighting this!
> ###### Description of crowdsourcing
> * We will move the description into the main text for the final draft!
> ###### Details of Model setup
> * The model set up is reported in detail in the appendix (section A.1) and we will move this into the main text for the final draft. We aim to release our code and artifacts with the final version of the paper to allow for reproducibility. For additional transparency, given the frequent changes in models, we will also provide timestamps of when the models were prompted. We used the davinci model from Feb 2023 on all four datasets. Experiments on ada, babbage, curie, text-davinci, and GPT4 models on the TriviaQA dataset were all conducted in June of 2023. Experiments on ada, babbage, curie, text-davinci, and GPT4 models on the CountryQA, NaturalQA, and Jeopardy dataset were conducted last week, August 2023.
> ###### Evaluation design
> * We’ve used the time during the response period to evaluate all six models across the four datasets. We present the aggregated results across the four models below and will include detailed results for each model and dataset in the appendix. The results replicated almost exactly with the four datasets together, highlighting the potential harm of overconfidence and factive verbs as well as the gains of evidential markers.
>
> |          |Ada    |Babbage |Curie   |Davinci |text-davinci-003|GPT4    |
> |-----------------|-------|--------|--------|--------|----------------|--------|
> |Booster          |0.091  |0.257   |0.313|0.392   |0.589           |0.793   |
> |Hedge            |0.079  |0.272   |0.333***|0.468***|0.642***        |0.822***|
> |Evidentials      |0.087**|0.281***|0.347***|0.449*  |0.640***        |0.820***|
> |Not Evidentials  |0.080  |0.250   |0.301   |0.433   |0.601           |0.799   |
> |Factive Verbs    |0.078  |0.237   |0.293   |0.347   |0.555           |0.771   |
> |Not Factive Verbs|0.085* |0.276***|0.336***|0.468***|0.641***        |0.821***|
> t-test p-values, * < 0.05, ** < 0.01, *** < 0.001
>
> ###### Conclusiveness of the results from 4.2
> * Using our results with six models across all four datasets, we aggregated the results by template and using t-tests of independence, we identified six templates which perform significantly better (p-value < 0.05) than the standard method. We agree that stronger claims about this phenomenon would require further work, and we hope that the community will continue to investigate how epistemic markers can improve model accuracies.
> ###### Numerical test set was downsampled
> * The test sets were downsampled because introducing numerical prompting significantly increases the prompting budget we need. For each template, we  prompted the model seven times (once for each numerical interval), given our constraints, we felt it necessary to downsample the test sets.
> ######  “twice as often” should obviously be replaced by “half as often”.
> * Thank you for this! It should be “half as often”.
> ###### Refrain from using approximators
> * Great question! Approximators are great for describing items/things that are continuous. However, given that we are mainly working with trivia questions which have discrete answers (such as the capital of countries) we did not find approximators to be as relevant. We will include this justification in our final version. We would love to see work on this in the future!
> ###### Making datasets available:
> * We can definitely make the CountryQA dataset available to the community! The second dataset, Jeopardy, is not ours but is linked in our paper and we would encourage the community to use the link and cite it appropriately.
> We’ve included the list of templates in the appendix but can include it in our GitHub Repo as well.
> ###### Typos:
> Thank you for all these notes! We really appreciate the time you put into this review and will make sure to incorporate the feedback into our final version.

---

### Meta-Review · Area_Chair_tYFA · 2023-09-16

**Recommendation:** 5

**Metareview:**

This paper examines how LLMs respond to epistemic markers of certainty and uncertainty and develops a wide range of different modifiers based on linguistic theory, which are inserted into QA prompts. The authors conduct experiments which turn out to contradict the hypotheses/expectations (i.e., LLMs are robust to the modifiers and the certainty of the modifiers increases accuracy and vice versa). The reviewers find this paper very intriguing to read and that analyzing the behavior of LLMs w.r.t. epistemic markers of uncertainty and specific linguistic phenomena in general is valuable to the community and the paper has conducted comprehensive experiments and analysis. Most concerns have been addressed in the rebuttal and the authors promised to incorporate the feedback into the revision. Overall, I'd recommend accepting this paper to the main conference.

---

### Decision · Program_Chairs · 2023-10-07

**Decision:**

Accept-Main

**Comment:**

This paper examines how LLMs respond to epistemic markers of certainty and uncertainty and develops a wide range of different modifiers based on linguistic theory, which are inserted into QA prompts. The authors conduct experiments which turn out to contradict the hypotheses/expectations (i.e., LLMs are robust to the modifiers and the certainty of the modifiers increases accuracy and vice versa). The reviewers find this paper very intriguing to read and that analyzing the behavior of LLMs w.r.t. epistemic markers of uncertainty and specific linguistic phenomena in general is valuable to the community and the paper has conducted comprehensive experiments and analysis. Most concerns have been addressed in the rebuttal and the authors promised to incorporate the feedback into the revision. Overall, I'd recommend accepting this paper to the main conference.